# Comparison of Performance in the Six-Minute Walk Test (6MWT) between Overweight/Obese and Normal-Weight Children and Association with Haemodynamic Parameters: A Cross-Sectional Study in Four Primary Schools

**DOI:** 10.3390/nu16030356

**Published:** 2024-01-25

**Authors:** Alice Giontella, Angela Tagetti, Sara Bonafini, Denise Marcon, Filippo Cattazzo, Irene Bresadola, Franco Antoniazzi, Rossella Gaudino, Paolo Cavarzere, Martina Montagnana, Angelo Pietrobelli, Claudio Maffeis, Pietro Minuz, Cristiano Fava

**Affiliations:** 1Department of Medicine, University of Verona, 37100 Verona, Italy; alice.giontella@gmail.com (A.G.); angela.tagetti@libero.it (A.T.); denise.m@hotmail.it (D.M.); cattazzof@gmail.com (F.C.); irene.bresadola@aovr.veneto.it (I.B.); pietro.minuz@univr.it (P.M.); cristiano.fava@univr.it (C.F.); 2Department of Surgical, Odontostomatological and Maternal-Infantile Sciences, University of Verona, 37129 Verona, Italy; franco.antoniazzi@univr.it (F.A.); rossella.gaudino@univr.it (R.G.); paolo.cavarzere@aovr.veneto.it (P.C.); angelo.pietrobelli@univr.it (A.P.); claudio.maffeis@univr.it (C.M.); 3Department of Engineering for Innovative Medicine, University of Verona, 37100 Verona, Italy; martina.montagnana@univr.it; 4Pennington Biomedical Research Center, Baton Rouge, LA 70808, USA

**Keywords:** obesity, children, physical activity, Rate Pulse Product, six-minute walk test, blood pressure, heart rate

## Abstract

Physical activity plays a pivotal role in preventing obesity and cardiovascular risks. The six-minute walk test (6MWT) is a tool to assess functional capacity and predict cardiovascular events. The aim of this cross-sectional study was to compare the performance and haemodynamic parameters before and after a 6MWT between obese/overweight vs. normal-weight children (average age 8.7 ± 0.7 years) participating in a project involving four primary schools in South Verona (Italy). Validated questionnaires for physical activity and diet, as well as blood drops, were collected. Overweight or obese children (OW&OB; *n* = 100) covered a shorter 6MWT distance compared to normal-weight children (NW, *n* = 194). At the test’s conclusion, the OW&OB group exhibited a higher Rate Pulse Product (RPP = Systolic Blood Pressure × Heart Rate) as compared to the NW. Body Mass Index, waist-to-height ratio, fat mass by electrical impedance, and trans fatty acids showed direct correlations with pre and post-test haemodynamic parameters, such as RPP, and inverse correlations with oxygen saturation. OW&OB children demonstrated lower performance in this low-intensity exercise test, along with an elevated haemodynamic response. Excess fat in childhood can be considered a risk factor for haemodynamic stress, with potential deleterious consequences later in life. Efforts should be initiated early to break this cycle.

## 1. Introduction

Childhood obesity has become a global epidemic, significantly impacting cardiovascular health in children and adolescents [1]. In Europe, the proportion of children 7–9 years old with overweight (OW) and obesity (OB) is around 30%, with a modest trend toward a decrement, especially in southern countries in recent years [2], whereas in the US the figure is even worse [3].

Childhood and adolescent obesity are associated with established risk factors for cardiovascular diseases, including elevated systolic and diastolic blood pressure (SBP and DBP), atherogenic dyslipidemia, atherosclerosis, metabolic syndrome, type II diabetes mellitus, and vascular and cardiac structural and functional changes [4,5,6].

The impact of an incorrect diet and sedentary lifestyle on children’s health has been studied extensively [7,8]. 

Several epidemiological studies showed a correlation between consumption of unhealthy foods and adiposity, and a recent meta-analysis suggests that the intake of sugar-sweetened beverages (SSBs) and unhealthy foods during childhood might lead to an elevation in BMI/BMI Z-Score, percentage body fat, or the likelihood of overweight/obesity [9].

Regarding physical activity, a study among first-year primary school children in Modena (Italy) found that 75% spent less than 7 h per week in physical exercise, while 63.9% engaged in sedentary activities for two or more hours daily. Excessive screen time, including tablet, computer, phone, and video game use, significantly contributed to childhood overweight/obesity [7]. 

Interventions promoting physical activity and reducing sedentary behaviours are considered crucial for addressing childhood obesity and improving overall health outcomes in children, even if long-term studies on hard endpoints are lacking [10]. Efforts to encourage healthy lifestyle changes for youth are imperative to combat the rise in childhood obesity and promote better health.

The six-minute walk test (6MWT) is a sub-maximal exercise test used to assess aerobic exercise capacity and functional performance. It measures the distance an individual can walk in six minutes and is commonly used to assess the impact of various conditions on physical function [11].

It has demonstrated significant prognostic value in various medical conditions, such as chronic heart failure [12] or primary pulmonary hypertension [13]. The 6MWT is widely utilised, even in school-aged children [14,15,16]. It has also been employed in obese patients, where it is associated with body composition [17,18] and performance after a weight reduction programme [19] or bariatric surgery [20]. Even in obese children, 6MWT showed good reproducibility [21]. Interestingly, the 6MWT distance (6MWT-D) performed by obese children was, on average, 14% less than the distance that normal-weight children walked [21].

In haemodynamics, the Rate Pressure Product (RPP) is an index of myocardial oxygen consumption and is proportional to the workload of the heart. It is calculated by multiplying the systolic blood pressure (SBP) by the heart rate (HR). This measure is used to estimate the myocardial (heart muscle) work and reflects the energy consumption of the myocardial muscle. It has been shown to have prognostic value in various adult populations, including in predicting cardiovascular events and mortality [22,23].

In recent years, we presented the results of a survey that was conducted in four primary schools in South Verona, where the percentage of obese and overweight children was 21.3% and 13.0%, respectively [24].

On the same occasion, we invited willing children to participate in a 6MWT, during which we measured HR and BP both before and after the test. The results obtained from normal-weight (NW) and overweight/obese (OW&OB) children were recorded and analysed. 

We hypothesised that OW&OB children may exhibit poorer performance and an impaired haemodynamic profile during exercise compared to normal-weight (NW) children. Thus, the primary objective of this study was to assess the differences in walked distance (6MWT-D) and haemodynamic parameters (particularly BP and the RPP) following the 6MWT between NW and OW&OB children. Subsequently, we explored potential associations between these measures and (i) fat disposition estimated by impedance; (ii) glucose, lipids, and fatty acids in a blood drop sample; and (iii) food intake and physical activity patterns documented through validated questionnaires.

## 2. Materials and Methods

A detailed description of the collection of the samples and tests performed is reported elsewhere [24].

### 2.1. Population Selection

In summary, all children from the third and fourth grades of four primary schools in the Verona South district were invited to participate. A brief presentation of the project was given to both the children and their parents approximately one week before the study day. The only exclusion criteria were the child’s unwillingness to participate or the absence of written parental consent. The Verona South district was selected based on the collaboration of the headmasters. The study, approved by the Ethical Committee of Verona and Rovigo (CESC *n* = 375, approval date: 18 February 2015), followed a cross-sectional observational design with all measurements conducted on a single morning, and one day was designated for each school. Written informed consent was obtained from all children’s parents. 

Five researchers were engaged in collecting all the measurements: three from the Internal Medicine Division of the Hypertension Excellence Centre of the European Society of Hypertension in Verona, who concentrated on haemodynamic parameters, and two paediatricians responsible for anthropometric measurements and sample collection. Additionally, one dietitian was responsible for gathering questionnaires (see below).

### 2.2. Anthropometric and Haemodynamic Measurements

Anthropometric measurements were taken in the school gym in the morning, with children wearing light clothing and no shoes. Weight and height were measured using a calibrated balance and stadiometer; the waist was measured using a flexible, non-stretchable measuring tape; and Body Mass Index (BMI) was calculated as weight over height^2^ (kg/m^2^), whereas Body Surface Area (BSA) was measured using the Monteller formula [BSA (m^2^) = ([Height (cm) × Weight (kg)]/3600)^1/2^]. Children were classified as overweight (≥85th and <95th percentile for age and sex) or obese (≥95th percentile for age and sex) based on BMI percentiles (WHO child growth standard) [25,26]. Body composition was assessed using bioelectrical impedance analysis (Tanita MC 780 MA, Tanita Corporation, Tokyo, Japan). Specifically, the estimation included fat percentage (%), fat mass (FM in kg), fat-free mass (FFM in kg), total body water (TBW in kg), and basal metabolic rate (BMR in kJ and kcal). 

Rest brachial blood pressure (BP) was determined as the average of three measurements in the supine position, whereas children were lying on a gym mat using a validated for children semiautomatic oscillometric device (Omron 750 IT) [27] that also recorded heart rate (HR) at baseline. Within the first minute after the completion of the 6MWT, BP and HR were measured twice using the same device as before in the supine position. The RPP was calculated by multiplying the SBP by the HR.

BP measurements were expressed even as a Z-Score, and percentiles were determined according to established guidelines [28].

### 2.3. 6MWT

Willing children were invited to perform the 6MWT while they were in the gym after the baseline haemodynamic tests, wearing light clothing. During the test, the children were invited to walk at their normal pace for six minutes, aiming to cover as much distance as possible. At the end of the 6MWT, the total distance covered (6MWT-D, m) was recorded and haemodynamic parameters measured, as specified above [16]. 

### 2.4. Laboratory Measurements

Nearly at noon, following a minimum 4 h fasting period, a small amount of blood was voluntarily obtained from children through a fingerprick for the measurement of plasma cholesterol, triglycerides, and glucose. This was achieved using two point-of-care testing (POCT) devices: HPS Multicare-in from Biochemical System International, Arezzo, Italy, for cholesterol and triglycerides, and Nova Biomedical, Waltham, MA, USA, for glucose [29,30]. For fatty acid analysis, a single drop of collected whole blood was directly applied to a filter paper (Ahlstrom 226, PerkinElmer, Greenville, SC, USA) that had been pre-treated with an antioxidant cocktail (Oxystop, OmegaQuant Analytics, LLC., Sioux Falls, SD, USA) to safeguard unsaturated FAs from oxidation. Subsequently, the collected cards were promptly transported to Omegametrix GmbH (Martinsried, Germany) for analysis using capillary gas chromatography, as previously detailed [31]. Fatty acid levels are reported as a weight percentage of the total blood fatty acids (composition). The stability of FAs collected and stored in this manner has been assessed previously, and no sample degradation was identified [32,33].

### 2.5. Questionnaires

Additionally, two validated questionnaires, “Food Frequency Questionnaire” (FFQ) and “Physical Activity Questionnaire for Older Children” (PAQ-C), were administered. The questionnaires were presented to the children and their parents during a prior informative session, then completed at home with the parents and reviewed on the evaluation day by a dedicated dietician.

### 2.6. Food Frequency Questionnaire (FFQ)

A validated Food Frequency Questionnaire (FFQ) was used to assess the children’s typical consumption of 61 items, with responses recorded on a 5-point scale ranging from “never” to “more than once daily” [34]. The study aimed to determine the association of diet with diseases through various approaches, including investigating single FFQ intake, main food group intake, and dietary patterns in relation to other collected variables. Two dietary patterns, “healthy” and “unhealthy” ones, which were previously derived using Principal Components Analysis (PCA), were also associated with haemodynamic parameters after 6MTW. PCA is a widely used method for identifying behavioural patterns. These patterns provide insight into the usual combinations of individual foods or groups of foods and offer valuable information regarding the association between diet and disease. The “healthy” first principal component was represented by the high consumption of fish, legumes, vegetables, fresh fruits and nuts, and dairy products. The “unhealthy” pattern was then characterised by the consumption of cereals and tubers, sweets, fast food, meat, and eggs. More details are presented elsewhere [20].

### 2.7. Physical Activity Questionnaire

The Physical Activity Questionnaire (PAQ-C) is a seven-day recall consisting of nine statements about the frequency of physical activities at school, at home, and during leisure time [35]. Each item is assigned a score from 1 to 5, and a mean total score of physical activity, known as the PAQ-C score, is then calculated. While the PAQ-C provides a valid and reliable method to assess general levels of physical activity, it does not offer specific information about frequency, time, or intensity. To address this limitation, the first item of the PAQ-C has been supplemented with a semiquantitative question to define one’s physical activity level in terms of the Metabolic Equivalent of Task (MET). This approach yields a total of MET minutes/week, and specific thresholds are used to categorise subjects based on their adherence to moderate–vigorous physical activity.

### 2.8. Statistics

Data are expressed as the mean standard deviation (SD) for continuous variables or percentages for categorical ones. The level of the *p*-value was set at 0.05. The Spearman correlation coefficient. The Student’s *t*-test was utilised to compare variables between the two groups. The relationship between categorical data was examined using the chi-square test. Multivariate linear regression models were conducted to assess the association of diet patterns and/or physical activity with the results of the 6MWT. The statistical analyses were performed using SPSS (version 23; IBM Corp., Armonk, NY, USA) and R (version 4.2.0 R Foundation for Statistical Computing), while the graphs were created with GraphPad Prism version 7.00 for Windows.

## 3. Results

A total of 309 out of 413 children belonging to the third and fourth classes of the four schools, constituting a participation rate of 74.8%, took part in the main study, with 294 (95%) expressing willingness to complete the 6MWT as well. The baseline characteristics of the participants, categorised into NW and OW&OB, are comprehensively outlined in Table 1.

The 6MWT-D was found to be lower in the OW&OB group. As anticipated, all body size and disposition measures were higher in the OW&OB group, including SBP and the RPP, before and after the 6MWT, while HR was higher only after the exercise. In Table 2 and Figure 1, correlations between adiposity, body composition measurements, and performance, along with haemodynamic parameters after the 6MWT, are presented. Various measures, particularly BMI, fatty mass (FM), and waist-to-height ratio, exhibited direct correlations with pre- and post-test haemodynamic parameters, such as RPP, and inverse correlations with oxygen saturation.

Table 3 displays the relationship between food intake/physical activity patterns and outcome variables. 

In an additional exploratory analysis (Table 4), correlations between fatty acids, glucose, and triglycerides collected from a blood drop with post-6MWT-D and haemodynamic parameters were examined. Only Trans-FAs exhibited a weak association with post-exercise SBP, RPP, and 6MWT-D. Omega-3 FAs were associated with 6MWT-D.

These results remained largely unchanged when considering the increase in haemodynamic parameters (Delta-SBP, Delta-DBP, Delta-HR, and Delta-RPP) as the target variables (see Appendix A).

### Linear Regressions

After adjustment for sex, age, and height, most of the relationships between BMI, FAT, FFM, FFM/FM mass, and all the haemodynamic parameters measured post-6MWT, previously identified through bivariate correlations, were confirmed (highlighted in bold in Table 2, Table 3 and Table 4). Similarly, the associations initially observed between Trans-FA and SBP and RPP, as well as between Trans-FA and Omega-3 FA with 6MWT walking distance, persisted in the adjusted linear regression model. Linear regressions are included in the Appendix A.

## 4. Discussion

The main findings of this study are that, even in childhood (ages 7–10), individuals classified as overweight or obese (OW&OB) exhibit poorer performance in the 6MWT compared to their normal weight (NW) counterparts. Additionally, their haemodynamic parameters are compromised both before and after the exercise. Furthermore, some measures of adiposity (or fat distribution) show a negative association with walking distance and a positive association with haemodynamics, suggesting an increased cardiac workload to achieve lower performance. It is physiologically accurate to say that BP and HR increase during physical exercise and remain elevated for a few minutes post-exercise. This is because the body’s demand for oxygen rises during exercise, leading to an increase in cardiac output achieved through elevated HR and stroke volume. This surge in cardiac output, along with a transient rise in systemic vascular resistance, particularly elevates SBP [36], probably through an enhanced Sympathetic Nervous System response.

However, it is crucial that this increase stays within specific physiological limits. For instance, current guidelines in sports medicine certify that SBP should not exceed 220 mmHg during and after exercise to meet eligibility criteria for competitive sports [37].

Despite this, the clinical and prognostic significance of the rise in RPP, HR, and BP after exercise remains unclear, both in adults and children. Recent guidelines from the European Society of Hypertension caution against the routine use of exercise testing as an assessment for hypertension. This is due to various constraints, including the absence of standardised methodology and definitions [38].

However, a meta-analysis indicates that the exercise-induced hypertensive response, particularly at moderate exercise intensity during exercise stress testing, can predict cardiovascular events and mortality [39]. This is likely attributed to the excessive pulsatile effect that this increase in BP and HR can impart to distal arterioles and capillaries, contributing to hypertension-mediated end-organ damage [40].

It is important to note that the RPP can offer valuable insights into the workload of the heart and should be critically assessed when evaluating the cardiovascular health of patients. Particularly in young OW&OB children, the increase in workload after minimal effort (during the 6MWT, children were invited to walk at their own pace) could be an indicator of muscle deconditioning, easy fatigability, and potentially increased cardiovascular stress.

All these factors, on one side, can contribute to a vicious cycle that discourages young OW&OB children from engaging in more physical activity, potentially worsening their cardiovascular health. On the other side, the repeated stress imposed on the heart and arteries, even during mild exercise, could contribute to initiating a process of gradual cardiac and vascular hypertrophy [4,41].

Regarding factors associated with a decrease in the 6MWT-D, we want to draw attention to several measures of adiposity, partly estimated by bioelectrical impedance, which showed significant associations in our sample: BMI, waist/height ratio, BSA, FM, as well as FFM and TBW.

Several studies have explored the correlation between fat measures and performance during the 6MWT. A study involving 90 obese patients revealed a positive relationship between the distance walked during the 6MWT and fat-free mass (FFM) in both men and women [17].

A study conducted on obese girls found that BMI, body height, and fat-free mass (FFM) all exhibit correlations with the 6MWT distance. The study concludes that assessing performance based on body composition could facilitate the comparison of 6MWT distance data between obese individuals and age- and gender-matched normal-weight children [18]. This finding is in line with similar observations made in middle-aged and older healthy adults [42].

However, beyond the distance covered, the relationships between adiposity measures and haemodynamic parameters are particularly relevant. This aspect is only partially supported by other observational studies. Twenty-one overweight or obese adults (average BMI exceeding 37 kg/m^2^) underwent the 6MWT twice on the same day. The results revealed a negative correlation between walking distance and BMI (r = −0.47, *p* = 0.03) and pre-test heart rate (r = −0.54, *p* = 0.01), suggesting the potential utility of 6MWT as a fitness indicator in both clinical studies and healthcare practices for this population [43].

Other studies have analysed BP, HR, cardiac output, stroke volume, and pulmonary vascular resistance, but in selected groups of patients, such as those with pulmonary artery hypertension or aortic stenosis, making comparisons with our study challenging [44].

It is important to note that the 6MWT is not the sole validated tool for assessing fitness in both children and adults. Submaximal or maximal treadmill tests are widely used in both clinical and research settings.

In a study involving 30 obese and 30 NW males (age range 18–45 years), participants underwent a submaximal treadmill exercise. Resting and post-exercise HR, SBP, and DBP were significantly higher and remained elevated in the obese group compared to the non-obese group [45].

Ten non-obese and ten obese men underwent submaximal exercise involving stepwise incremental cycling until reaching 60% of their age-predicted maximum HR. The baseline HR was higher in the obese group with slower recovery, while cardiac output and RPP were elevated post-exercise [46].

A study conducted on seemingly healthy OW men revealed impaired post-exercise haemodynamic responses. Specifically, OW individuals exhibited higher SBP, with selectively increased DBP persisting up to 60 min following exercise compared to normal-weight individuals [47].

In another investigation, forty-six NW and twenty-one OB participants underwent measurements of peripheral and central BP at rest, as well as 15 and 30 min following acute maximal exercise. OB individuals exhibited an overall higher central and peripheral BP without any exercise-induced effects. Additionally, the increase in HR post-exercise was greater in obese individuals compared to their NW counterparts [48].

In a Chinese study involving 940 male participants (average age 36.8 years, average BMI 26 kg/m^2^) who underwent a two-stage load test on cycle ergometry, the results indicated that at the end of the exercise, HR increased in all quintiles of body fat percentage, while BP did not show a significant increase [49]. 

Thus, the present study further supports the evidence that obesity, particularly fat, diminishes aerobic performance and puts stress on haemodynamics. It demonstrates that this imbalance is present early in childhood.

We also investigated a potential association with dietary and physical activity patterns obtained from validated questionnaires, but we could not identify any significant associations. Similarly, there was no significant association found between blood fatty acids, glucose, and triglycerides obtained from a blood sample collected on the same day as the 6MWT.

Interestingly, a study conducted on Chilean children revealed a significant association between insulin and HOMA, but not triglycerides or glucose levels, with ΔHRR, especially among overweight and obese children [50].

Regarding fatty acids (FA), while some studies have suggested a potential beneficial effect of omega-3, especially eicosapentaenoic acid and docosahexaenoic acid, on performance, such as improved endurance capacity, delayed onset of muscle soreness, and markers related to enhanced recovery and immune modulation in competitive or amateur athletes, the issue remains open to debate [51]. The weak correlation we found between omega-3 and 6MWT-D should be approached with caution. Similarly, the very weak association we observed between Trans-FA—a type of unsaturated fat found in processed foods—and palmitic acid, pre- and post-exercise systolic blood pressure (SBP), and post-6MWT Rate Pressure Product (RPP)—although potentially intriguing—warrants replication in independent samples before further consideration. It is worth noting that the consumption of trans fats has been linked to an increased risk of heart disease and other health problems, as they elevate LDL-cholesterol and lower HDL-cholesterol levels [52].

### Limitations of the Study

Our study has several limitations. Firstly, the sample is considerably “convenience-based”, and its size is relatively limited. The generalizability of this to children from different locations and age groups may be limited. The cross-sectional design and the observational nature of the study prevent drawing any conclusions about causal links among the observed associations. Additionally, data on physical activity and food intake were collected solely through questionnaires, which may be susceptible to misreporting. The assessment of fatty acids, glucose, and lipids from a blood drop had a fasting period of only 4 h.

On the positive side, we utilised validated devices to measure blood pressure, and laboratory parameters in children were assessed with rigour. We also made efforts to maximise the accuracy of the information gathered from questionnaires by involving a dietitian who thoroughly reviewed them with all the children.

## 5. Conclusions

In conclusion, young children with overweight or obesity displayed elevated baseline cardiovascular parameters and exhibited heightened responses to moderate exercise, possibly indicating altered autonomic functions characterised by sympathetic hyperactivity. Consequently, despite a poorer performance compared to their NW counterparts, these children experienced an increased workload. Although the long-term prognostic significance of these findings is challenging to predict, it is concerning that haemodynamic parameters, particularly SBP, tend to be higher in this “at-risk” group of children. This heightened state could make even low-intensity exercise more challenging for them. Civil and scientific societies should continue to make every possible effort to break this vicious circle, beginning early in childhood.

## Figures and Tables

**Figure 1 nutrients-16-00356-f001:**
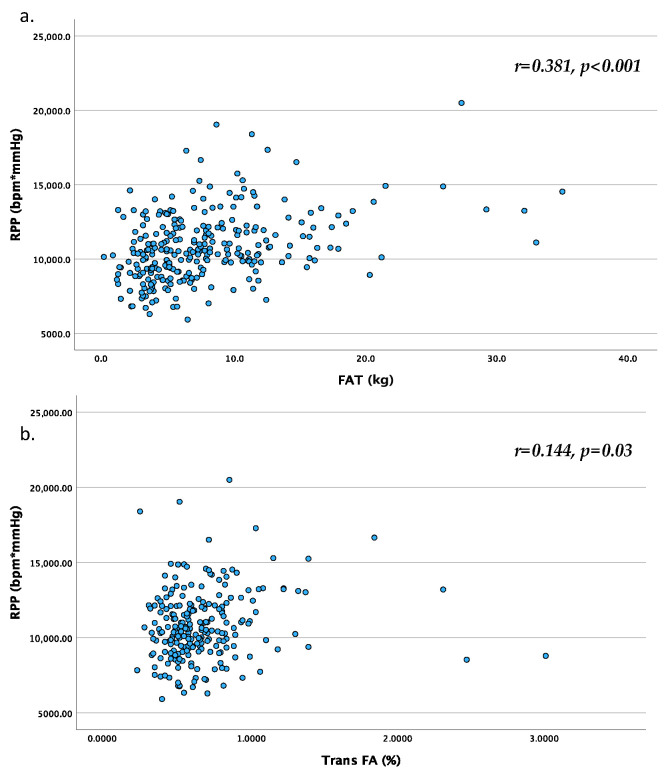
Graphical representation of the correlation between RPP and FAT (**a**) as well as Trans FA (**b**). Legend—RPP: Rate Pressure Product; FA: Fatty Acids.

**Table 1 nutrients-16-00356-t001:** General characteristics of the 294 children who participated in the 6MWT.

Characteristics	Normal-Weight (NW) *n* = 194	Overweight/Obese (OW&OB) *n* = 100	*p*-Value
Age, yr	8.7 ± 0.7	8.6 ± 0.7	0.143
Male, %	48.5	53.0	
Anthropometric and Body Composition (via impedenzometry)			
BMI, kg/m^2^	16.30 ± 1.7	21.8 ± 3.1	<0.001
BMI, percentile for age	46.7 ± 26.2	93.6 ± 3.8	<0.001
Waist/height	0.43 ± 0.06	0.51 ± 0.08	<0.001
Waist/hip	0.85 ± 0.08	1.00 ± 0.96	<0.001
BSA, m^2^	1.06 ± 0.11	1.27 ± 0.14	<0.001
FAT mass, kg	15.7 ± 5.1	12.9 ± 6.1	<0.001
FAT-FREE mass, kg	37.5 ± 25.0	29.5 ± 5.2	<0.001
FFM/FM	31.3 ± 6.3	2.7 ± 1.1	<0.001
TBW, kg	27.5 ± 18.4	21.9 ± 3.1	<0.001
6MWT distance, m	554.6 ± 2.6	535.7 ± 53.8	0.007
Pre-6MWT Haemodynamic Measurements			
SBP, mmHg	109.1 ± 10.5	113.4 ± 8.2	<0.001
z-SBP	0.86 ± 0.91	1.13 ± 0.76	0.010
DBP, mmHg	66.1 ± 7.8	67.8 ± 7.8	0.092
z-DBP	0.64 ± 0.71	0.72 ± 0.76	0.316
HR, bpm	80.3 ± 11.5	80.6 ± 11.8	0.818
RPP, mmHg * bpm	8797.6 ± 1750.3	9157.4 ± 1657.3	0.043
SatO_2_	98.8 ± 0.60	98.6 ± 1.5	0.228
Post-6MWT Haemodynamic Measurements			
SBP, mmHg	119.3 ± 12.2	126.4 ± 11.1	<0.001
z-SBP	1.8 ± 1.1	2.4 ± 1.0	<0.001
DBP, mmHg	68.2 ± 7.0	69.6 ± 7.8	0.143
z-DBP	0.81 ± 0.63	0.89 ± 0.70	0.414
HR, bpm	86.6 ± 13.0	95.4 ± 17.2	0.004
RPP, mmHg * bpm	10,366.5 ± 2103.4	11,593.0 ± 2405.3	<0.001
SatO_2_ %	98.8 ± 0.44	98.4 ± 0.9	<0.001
Delta (Post–Pre-6MWT) Haemodynamic Measurements			
Delta PAS	10.15 ± 9.80	13.0 ± 11.17	0.032
Delta PAD	1.09 ± 8.32	1.85 ± 7.45	0.792
Delta HR	6.32 ± 12.14	10.76 ± 11.56	0.003
Delta RPP	1574.5 ± 1927.4	2436.3 ± 2042.0	<0.001
Delta SatO_2_	0.05 ± 0.44	0.17 ± 1.58	0.519
Physical Activity (from the PAQ-C Questionnaire)			
Sedentary activity, MET * min/week	3591.2 ± 3919	2968.7 ± 4154.1	0.236
Intense physical activity, MET * min/week	1606.3 ± 1367.8	1766.2 ± 2697 ± 0	0.591

Legend—*: multiplication; 6MWT: Six-Minute Walk Test; BMI: Body Mass Index; BSA: Body Surface Area; DBP: Diastolic Blood Pressure; HR: Heart Rate; RPP: Rate Pressure Product; SBP: Systolic Blood Pressure; TBW: Total Body Water; and SatO_2_: Saturation of Oxygen.

**Table 2 nutrients-16-00356-t002:** Correlations between body composition parameters also derived from bioelectrical impedance analysis and haemodynamic parameters measured after the 6MWT.

	Post-6MWT SBP	Post-6MWT Z-SBP	Post-6MWT DBP	Post-6MWT Z-DBP	Post-6MWT HR	Post-6MWT RPP	Post-6MWT SatO_2_	6MWT Distance
BMI, kg/m^2^	**0.361 ****	**0.318 ****	**0.209 ****	**0.145 ***	**0.228 ****	**0.345 ****	**−0.313 ****	**−0.212 ****
z-BMI	**0.324 ****	**0.291 ****	**0.161 ****	**0.123 ***	**0.189 ****	**0.294 ****	**−0.267 ****	**−0.192 ****
Waist/height	**0.175 ****	**0.160 ****	**0.120 ***	0.094	**0.150***	**0.193 ****	0.081	−0.089
Waist/hip	0.023	0.012	−0.09	−0.006	0.017	0.018	0.128	−0.058
BSA, m^2^	**0.480 ****	**0.352 ****	**0.205 ****	0.046	**0.165 ****	**0.354 ****	**−0.234 ****	**−0.120 ***
FAT, kg	**0.393 ****	**0.163 ****	**0.241 ****	**0.163 ****	**0.252 ****	**0.381 ****	**−0.354 ****	**−0.194 ****
FFM, kg	**0.375 ****	**0.247 ****	0.099	−0.047	0.062	**0.236 ****	−0.034	−0.036
FFM/FM	**−0.305 ****	**−0.288 ****	−0.157 **	**−0.120 ***	−0.113	**−0.232 ****	0.149	0.114
TBW, kg	**0.402 ****	**0.262 ****	0.094	−0.070	0.060	**0.239 ****	−0.088	−0.071

Legend—6MWT, Six-Minute Walk Test; BMI: Body Mass Index; BSA: Body Surface Area; DBP: Diastolic Blood Pressure; FM: Fat Mass; FFM: Fat-Free Mass; HR: Heart Rate; RPP: Rate Pressure Product; SBP: Systolic Blood Pressure; and TBW: Total Body Water. *: *p*-value < 0.05; **: *p*-value < 0.01. The correlations that remained significant after adjustment for confounding variables are highlighted in bold.

**Table 3 nutrients-16-00356-t003:** Correlations between PCA patterns/physical activity from specific questionnaires and haemodynamic parameters measured after the 6MWT.

	Post-6MWT SBP	Post-6MWT Z-SBP	Post-6MWT DBP	Post-6MWT Z-DBP	Post-6MWT HR	Post-6MWT RPP	Post-6MWT SatO_2_	6MWT Distance
**Healthy Pattern**	0.063	0.049	−0.002	−0.031	0.031	0.056	0.052	0.052
**Unhealthy Pattern**	0.066	0.047	0.067	0.054	0.009	0.029	−0.039	−0.039
**Sedentary Activity (MET * min/week)**	0.079	0.088	−0.039	−0.032	−0.038	0.009	0.083	0.083
**Intense Physical Activity (MET * min/week)**	−0.012	0.001	−0.081	−0.066	0.039	0.031	−0.007	−0.007

Legend—6MWT, Six-Minute Walk Test; DBP: Diastolic Blood Pressure; HR: Heart Rate; RPP: Rate Pressure Product; and SBP: Systolic Blood Pressure. *: *p*-value < 0.05.

**Table 4 nutrients-16-00356-t004:** Correlations between different metabolites and fatty acids from blood drops and haemodynamic parameters measured after the 6MWT in a subgroup of children (*n* = 232) with FA measurement.

	Post-6MWT SBP	Post-6MWT Z-SBP	Post-6MWT DBP	Post-6MWT Z-DBP	Post-6MWT HR	Post-6MWT RPP	Post-6MWT SatO_2_	6MWT Distance
**Omega 6 FA, %**	0.002	−0.016	−0.026	−0.033	0.028	0.019	−0.005	−0.059
**Omega 3 FA, %**	0.027	−0.023	0.111	0.089	0.016	0.019	−0.043	**0.159 ***
**Omega 9 FA, %**	0.061	0.098	−0.020	0.002	0.011	0.038	−0.121	−0.073
**Saturated FA, %**	−0.085	−0.084	−0.021	−0.019	−0.06	−0.088	0.098	0.055
**Trans FA, %**	**0.143 ***	**0.142 ***	0.232	0.031	0.101	**0.144 ***	−0.095	**0.140 ***
**Glucose, mg/dL**	0.061	0.050	−0.029	−0.030	−0.072	−0.031	−0.023	−0.033
**Triglycerides mg/dL**	0.111	0.115	**0.238 ****	**0.242 ****	0.080	0.112	**−0.250 ***	0.046
**Cholesterol mg/dL**	−0.135	−0.096	−0.071	0.003	−0.113	**−0.153 ***	0.197	0.066

Legend—6MWT, Six-Minute Walk Test; DBP: Diastolic Blood Pressure; FA: Fatty Acids; HR: Heart Rate; RPP: Rate Pressure Product; and SBP: Systolic Blood Pressure. *: *p*-value < 0.05; **: *p*-value < 0.01. The correlations that remained significant after adjustment for confounding variables are highlighted in bold.

## Data Availability

The data presented in this study are available on request from the corresponding author. The data are not publicly available due to privacy/ethical restrictions.

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
