# Peer review of "Comparison of Performance in the Six-Minute Walk Test (6MWT) between Overweight/Obese and Normal-Weight Children and Association with Haemodynamic Parameters: A Cross-Sectional Study in Four Primary Schools"

_nutrients, 2024, doi:10.3390/nu16030356_

Round 1
Reviewer 1 Report
Comments and Suggestions for Authors
I would like to congratulate the authors for their contribution to the area and for sending me this article for review.
This article entitled Comparison of Performance in the Six-Minute Walk Test 2 (6MWT) Between Overweight/Obese and Normal-Weight Chil-3 drain: Association with Hemodynamic Parameters attempts to compare a physical fitness test between children with and without overweight and obesity, an aspect that is considered interesting, however, it presents some shortcomings that will be discussed below.
The research design must be included in the title or summary.
The summary must be structured, although it does not include the titles introduction, objectives, method, results and conclusion. The target is not found.
It is recommended to check the punctuation and spacing between words.
A p value of 0.007 is included in the summary. It must be indicated and specified what this p value refers to. It is not understood.
In the summary it is indicated in the results section that a pre- and post-test was carried out, however, it has not been previously indicated in it that two measurements will be carried out.
It is not recommended to use the same words in the title and keywords.
The introduction includes numerous bibliography and interesting information, however the bibliography should be more current. As well as justify the use of this test in the school population and include quotes that use this test in this population. It is a widely used test and other tests are often used.
At the end of the introduction the objective should be included explicitly. Currently it is not written in the form of an objective.
It is recommended that the Strobe Statement be used to further detail all aspects of the method.
It is recommended to reorganize the method section. Currently the information is confusing.
The inclusion and exclusion criteria of participants are not indicated.
The conditions in which the subjects were measured should be described to a greater extent.
Indicate who carried out the measurements and number of researchers, information that was given to the participants, etc.
It is recommended to include appointments for the 6 minute walk test. As well as a reference for the food frequency questionnaire. An appointment for the PAQ-C questionnaire must also be included.
The results indicate a different sample than that indicated in the summary.
The results section shows data from two different moments of measurements, pre-post-test, however, this aspect is not mentioned throughout the article.
Reviewer 2 Report
Comments and Suggestions for Authors
The manuscript is interesting, however, I have the following comments.
1. In the introduction it is suggested to include that diet also influences body weight gain and the development of obesity or other chronic pathologies related to nutrition in children.
2. In the methodology it is necessary to include the number or registration of the bioethics protocol. Furthermore, it is necessary to improve the description of the inclusion and exclusion criteria.
3. Results. It is important that in the legends of the tables and figures, the authors are very specific about the origin of the results. For example, they correspond to blood tests or were obtained from the diet.
4. The authors performed FFQ, however the analysis of these results is very limited. I suggest further analysis, for example correlation between food group intake and hemodynamic parameters.
5. The discussion is good, but it needs to be improved. The authors should discuss more specific aspects regarding the results obtained and what physiological or pathophysiological processes would be involved.
II. Manor comments.
1. Improve the writing of the objective of the study.
2. Improve the writing of table legends.
3. Improve the resolution of the figure.
Round 2
Reviewer 1 Report
Comments and Suggestions for Authors
Congratulations
Reviewer 2 Report
Comments and Suggestions for Authors
Authors answered all my comments. Therefore, the manuscript can be accepted.